# Ionitronic manipulation of current-induced domain wall motion in synthetic antiferromagnets

Yicheng Guan[1,2], Xilin Zhou [1,2], Fan Li[1,2], Tianping Ma[1], See-Hun Yang[1] & Stuart S. P. Parkin [1✉]

The current induced motion of domain walls forms the basis of several advanced spintronic technologies. The most efficient domain wall motion is found in synthetic antiferromagnetic (SAF) structures that are composed of an upper and a lower ferromagnetic layer coupled antiferromagnetically via a thin ruthenium layer. The antiferromagnetic coupling gives rise to a giant exchange torque with which current moves domain walls at maximum velocities when the magnetic moments of the two layers are matched. Here we show that the velocity of domain walls in SAF nanowires can be reversibly tuned by several hundred m/s in a non-volatile manner by ionic liquid gating. Ionic liquid gating results in reversible changes in oxidation of the upper magnetic layer in the SAF over a wide gate-voltage window. This changes the delicate balance in the magnetic properties of the SAF and, thereby, results in large changes in the exchange coupling torque and the current-induced domain wall velocity. Furthermore, we demonstrate an example of an ionitronic-based spintronic switch as a component of a potential logic technology towards energy-efficient, all electrical, memory-in-logic.

[1] Max Planck Institute for Microstructure Physics, 06120 Halle, Germany. [2] These authors contributed equally: Yicheng Guan, Xilin Zhou, Fan Li. ✉email: stuart.parkin@mpi-halle.mpg.de

The current- or voltage-induced manipulation of magnetization is key to several spintronic technologies today, including magnetic random access memories (MRAMs)[1–3] and magnetic racetrack memories[4,5]. In the racetrack memory, a series of magnetic domain walls (DWs), which encode digital data, are manipulated in magnetic nanowires via current pulses[4,5]. These nanowires have evolved from in-plane[6] to out-of-plane[7] magnetized materials, to heterostructures with strong interfacial spin–orbit coupling[8,9], and, most recently, to synthetic antiferromagnetic (SAF) structures[10]. In the latter, the DWs can be moved very efficiently with velocities ($v$) that can exceed ~1 km/s by means of a giant exchange coupling torque (ECT) generated by the current[10]. Another means of manipulating magnetization are electric fields from gate voltages applied across insulating layers that have been shown, e.g., to tune the perpendicular magnetic anisotropy (PMA) of magnetic tunnel junctions and, thereby, allow for more energy-efficient MRAM[11–13]. To date, gate voltage-induced DW velocity changes in racetracks formed from ferromagnetic metals are very small due to the limited modifications of, e.g., the uniaxial anisotropy energy or interfacial Dzyaloshinskii Moriya interaction[14–17]. Here we show that, in a SAF structure, in which the lower and upper magnetic (LM and UM) layers are antiferromagnetically coupled via an ultrathin ruthenium layer[18,19], gate voltages applied through ionic liquids can significantly affect the DW velocity with changes of several hundred m/s. We show that this is due to an electrochemical process that is non-volatile but reversible and results from an enhancement or suppression of oxidation of the UM layer rather than from electrostatic changes in carrier density, as previously claimed[14–16,20–22]. Using these results, we demonstrate an ionic liquid gate (ILG)-controlled DW switch device.

## Results

### Manipulation of current-induced DW motion by ionic liquid gating.
SAF structures have the basic structure of UL/LM/Ru/UM/CL, where UL is an underlayer, LM and UM are magnetic sublayers that are coupled antiferromagnetically through a thin Ru layer, here 5 Å thick, unless otherwise stated; and CL is a capping layer. SAF structures were prepared by high vacuum magnetron direct current (DC) sputtering where UL = TaN(20)/Pt(15), LM = [Co(3)/Ni(7)/Co(1.5)], UM = [Co(5)/Ni(7)/Co(1.5)], and CL = TaN(15) (all thicknesses are in Å). Even though the individual layers are atomically thin, cross-section transmission electron microscopy imaging shows the layers are well defined (see Fig. 1a). Both the LM and UM layers display PMA, as shown via $M–H$ loops that are measured with an out-of-plane magnetic field (Fig. 2b). Nanowire structures (3 μm wide and 50 μm long) with lateral gate electrodes, as schematically outlined in Fig. 1b, were fabricated using optical lithographic techniques. Current-induced DW motion (CIDWM) measurements under the application of a gate voltage ($V_G$) applied across an ionic liquid to the SAF racetrack were carried out: typical sequences of four magneto-optical Kerr microscope images of CIDWM corresponding to the successive application of individual 10 ns-long pulses (current density $J = 1.2 \times 10^8 \, \text{A cm}^{-2}$) are shown in Fig. 1c ($V_G = 0$ V, pristine state).

The dependence of DW velocity $v$ on the injected current density under various $V_G$ is shown in Fig. 1d. First, when a positive $V_G$ is applied, the CIDWM efficiency monotonically increases. When a negative $V_G$ is subsequently applied, the CIDWM efficiency returns to its original value at −2 V and further degrades at −3 V. The dependence of $v$ on $V_G$ at a fixed $J = 1.2 \times 10^8 \, \text{A cm}^{-2}$ is shown in Fig. 1e. A very large change in $v$ of up to 130 m/s is found between $V_G$ of +4 V and −3 V in the SAF structure, which is much larger than previous reports of electric field-induced changes in DW motion for single

ferromagnetic layers[14–17]. By further increasing the current density, an even larger change in DW velocity of ~200 m/s can be achieved, as indicated in Fig. 1d. Interestingly, the threshold current density, below which the DWs do not move, is also slightly modified by $V_G$ (See Supplementary Fig. 1).

The nature of the coupling between the LM and UM layers can be inferred from the dependence of $v$ on a magnetic field $H_X$ that is applied in-plane along the racetrack (Fig. 1f). In these measurements, $H_X$ is varied from −900 to 900 Oe and $J$ is fixed at $1.2 \times 10^8 \, \text{A cm}^{-2}$. In the pristine state, the DW velocity shows a non-monotonic response to the application of $H_X$ for both up-down and down-up DW configurations (Fig. 1f and Supplementary Fig. 1), which is a distinct character of DWs in a SAF that are driven by an ECT[10]. For positive $V_G$ (+2 and +4 V), $v$ is highly sensitive to $H_X$ with pronounced peaks at distinct values of $H_X$. Such changes are indicative of larger contributions from the ECT. On the other hand, for negative $V_G$ (−2 V), the lower sensitivity of $v$ to $H_X$ provides evidence for a reduced ECT contribution. It is noteworthy that $v$ displays an almost linear response with $H_X$ for $V_G = −3$ V. Such a behavior is characteristic of the response expected from a simple ferromagnetic racetrack.

The reversibility of ionic liquid gating effects on CIDWM is demonstrated by comparing data after gate voltages of −3 V and 4 V are applied, as illustrated in Fig. 1g, h. The CIDWM at a given current density ($1.2 \times 10^8 \, \text{A cm}^{-2}$) robustly changes between a relatively low velocity of 110 m/s (−3 V) to a velocity of 240 m/s (at +4 V) that is more than two times higher (Fig. 1h). This experiment was repeated several times with the same results, clearly showing large reversible and non-volatile changes in the CIDWM induced by ionic liquid gating.

### Ionic liquid gating impact on magnetic properties.
The non-volatile nature of the ILG effect was further investigated using ex situ vibrating sample magnetometry (VSM) studies of the as-deposited films, as shown in Fig. 2a. The measured $M–H$ loops of samples using an out-of-plane magnetic field for a sequence of different $V_G$s are plotted in Fig. 2b. Typical SAF-like loops are observed for all $V_G$, except for −3 V. From these $M–H$ loops, $M_R/M_S$ and $M_S$ are extracted and summarized in Fig. 2c, where $M_R$ is the remnant magnetization at $H_Z = 0$ and $M_S$ is the saturation magnetization at $H_Z = 15$ kOe. As $V_G$ is increased from 0 to +4 V, $M_R/M_S$ gradually decreases from 0.113 to 0.0008, whereas negative $V_G$ results in an increase in $M_R/M_S$ to a value of 1 at $V_G = −3$ V. Meanwhile, $M_S t$, where $t$ is the total thickness of UM + LM, is enhanced from ~$1.20 \times 10^{-4} \, \text{emu cm}^{-2}$ in the pristine state to ~$1.33 \times 10^{-4} \, \text{emu cm}^{-2}$ for $V_G = +4$ V, but reduced to ~$0.666 \times 10^{-4} \, \text{emu cm}^{-2}$ for $V_G = −3$ V.

Thus, the effect of ILG on the CIDWM of SAF structures can be understood as follows: as the ECT-driven CIDWM in the SAF structure depends sensitively on the value of $M_R/M_S$[10], the reduced $M_R/M_S$ ratio by applying positive $V_G$ results in an increase in the ECT and, thus, an enhancement of CIDWM as observed. By contrast, negative $V_G$ suppresses $v$ due to a reduced ECT that results from an increase in the $M_R/M_S$ ratio. By further increasing the negative gate voltage, the SAF system is transformed into an single ferromagnetic (FM) system and a much reduced $v$ is observed, which results from the vanishing of the ECT in an FM system. The variation of the DW velocity $v$ vs. $M_R/M_S$, at a fixed current density $J = 1.2 \times 10^8 \, \text{A cm}^{-2}$, can be well described by a one-dimensional (1D) ECT analytical model, thereby substantiating these conclusions (see Supplementary Note 1 for details of 1D model).

### Electro-chemical mechanism of ionic liquid gating.
The non-volatile characteristic of the ILG effects on the magnetization of the SAF films indicates an electro-chemical origin of the ILG process. In

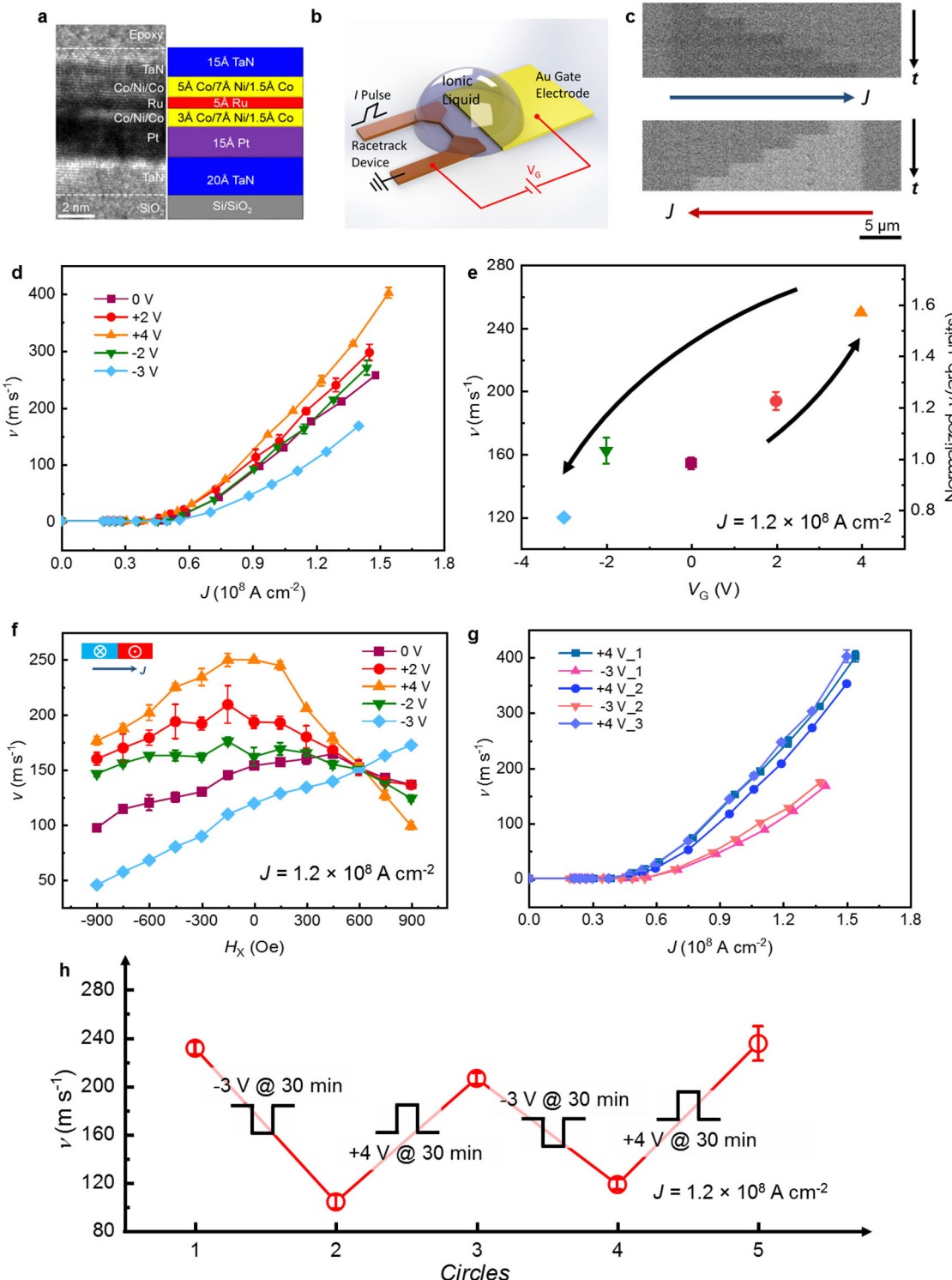

**Fig. 1 ILG effect on CIDWM in a SAF structure. a** Cross-section TEM image (left) and schematic structure (right) for a typical SAF structure. **b** Sketch of racetrack device with gate electrode. **c** Kerr microscopy images of a single DW moving along the nanowire at pristine state ($V_G = 0$ V): the images are taken after applying sequences of current pulses composed of $4 \times 10$ ns pulses (current density $J = 1.2 \times 10^8$ A cm$^{-2}$). Blue and red arrows indicate the current injection direction, and dark and bright contrast indicates up and down magnetization directions, respectively. **d** DW velocity $v$ as a function of current pulse density $J$ at different $V_G$. **e** $v$ at fixed $J = 1.2 \times 10^8$ A cm$^{-2}$ as a function of $V_G$, the black arrows indicate the sequence of application of $V_G$. **f** Dependence of $v$ on $H_x$ for various $V_G$, for down-up DWs at a fixed $J = 1.2 \times 10^8$ A cm$^{-2}$. **g** The reversibility of ILG effects on CIDWM of SAF structures is shown by varying the gating voltage from $-3$ V to $+4$ V several times and the corresponding DW velocity at a given current density of $1.2 \times 10^8$ A cm$^{-2}$ for different $V_G$ (**h**). The error bars in all figures correspond to 1 SD.

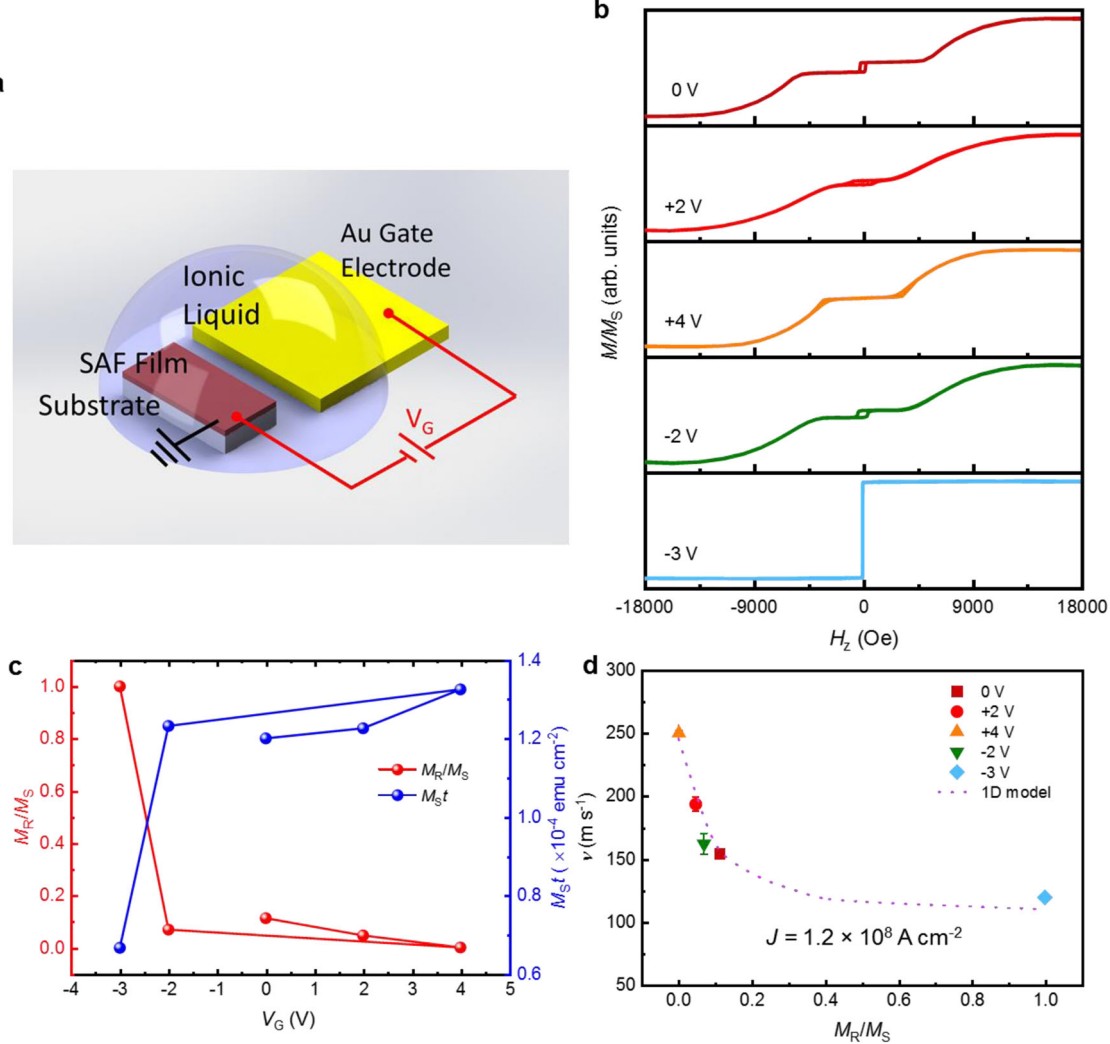

**Fig. 2 ILG-induced magnetization changes in SAF structures. a** Schematic illustration of ionic liquid gating of the pristine SAF films. **b** Magnetic hysteresis loops obtained from ex situ SQUID measurements for several $V_G$ applied in the following sequence: 0 V (pristine state), +2 V, +4 V, −2 V, and −3 V. **c** $V_G$ dependence of $M_R/M_S$ (left axis) and $M_St$ (right axis). **d** DW velocity at a fixed current density (1.2 × 10⁸ A cm⁻²) plotted as a function $M_R/M_S$. The dotted line corresponds to simulation results based on a 1D analytical model. The error bars in **d** correspond to 1 SD.

order to clarify this possibility, we studied the chemical nature of several SAF structures using ex situ X-ray photoelectron spectroscopy (XPS) for various $V_G$. As the XPS signal decays exponentially with depth from the surface of the film, the XPS spectra mainly arise from the UM and the TaN CL. Co-2$p$ and Ni-2$p$ core-level spectra are shown in Fig. 3a, b, respectively. The most important feature is the evolution of the satellite peaks at the Co-2$p_{3/2}$ and Ni-2$p_{3/2}$ edges. At the pristine state ($V_G = 0$ V), in addition to the dominant Co-2$p_{3/2}$ (778.2 eV) and Ni-2$p_{3/2}$ (852.6 eV) peaks, weak satellite peaks are observed at 780.7 eV for Co and 855.8 eV for Ni, which are attributed to a tiny degree of oxidation[23–25]. Such oxidation originates from oxygen that resides within the TaN CL (see Supplementary Fig. 3). Remarkably, $V_G = +4$ and $−3$ V suppresses and enhances, respectively, such satellite peaks compared to the pristine sample. This indicates a weakening and strengthening, respectively, of the oxidation of Co and Ni. The reversibility is also manifested by XPS characterization of a sample as the gate voltage was varied from 0 V to +4 V, to −3 V, and back to +4 V. The first data and the last taken at +4 V are almost the same, illustrating the robustness of the reversibility of the gating process.

Considering the experimental findings presented above, we propose the following mechanism for the effect of the ILG. The application of $V_G$ reversibly changes the oxidation state of the UM

layer and, thus, the magnetization of the UM. When the gate voltage reaches −3 V, the UM is heavily oxidized so that the SAF structure is effectively converted into an FM structure. The UM layer has a magnetization that is reduced by over 95%. Limited ionic liquid gate-induced oxidation of a single layer FM structure has been reported using HfO₂ CLs[17]. However, the ability to tune the CIDWM in SAF structures by ILG is much greater than that previously found for FM layers[15–17]. This is due to the fundamental differences in the primary mechanisms of CIDWM in FM and SAF structures, namely chiral-spin-orbit torque (SOT) in the former and ECT in the latter. Compared to chiral-SOT, the higher sensitivity of ECT to tiny changes in magnetic properties allows for a much greater impact of ILG in tuning the CIDWM and, thereby, gives rise to much larger changes in DW velocity.

**Ionic liquid gate-controlled racetrack logic switches.** We give two examples of how the large ILG effect on CIDWM that we have found for SAF racetracks can be used in a device. We first consider a "knot" racetrack device that consists of a straight wire that is separated into two curved lanes, as depicted in Fig. 4a. In the knot device, an injected DW is split into the two curved lanes and exits the lanes as two individual DWs. We find a distinctly

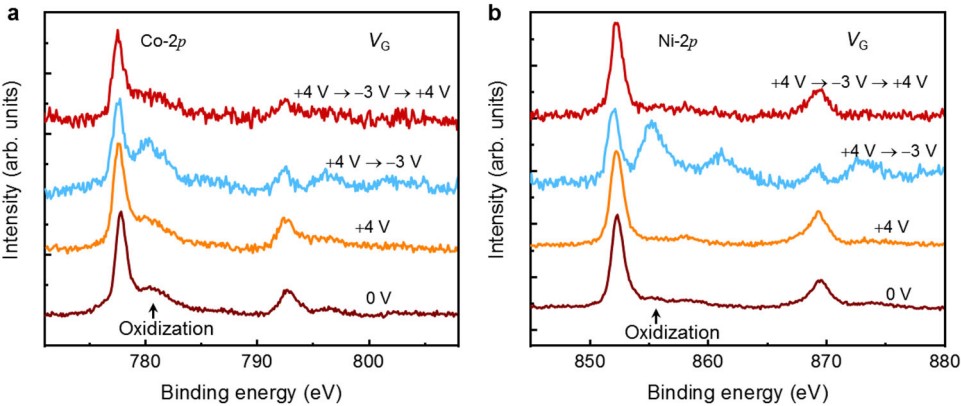

**Fig. 3 ILG-induced ion migration in SAF structures.** X-ray photoelectron spectra (XPS) of **a** Co-2$p$, **b** Ni-2$p$ for SAF samples gated at various $V_G$ applied in the following sequence: 0 V (pristine state), +4 V, +4 V → −3 V, and +4 V → −3 V → +4 V.

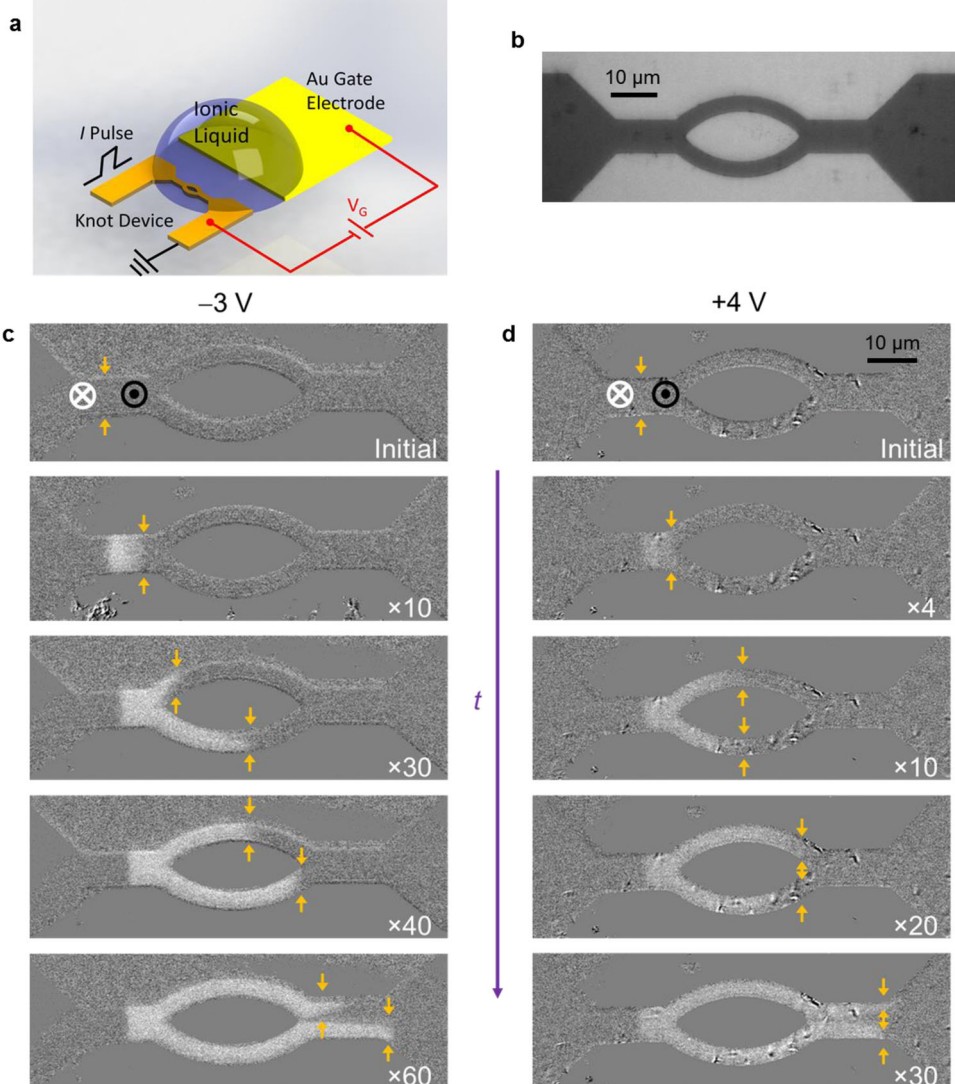

**Fig. 4 Knot devices controlled via ILG. a** Sketch of experimental setup for a knot device controlled by ionic liquid gating. **b** Optical micrograph of a typical knot device. Current-induced domain wall motion of a SAF knot device for $V_G = -3$ V (**c**) and +4 V (**d**). Current pulses with a magnitude and length of $1.2 \times 10^8$ A cm$^{-2}$ and 50 ns, respectively, were used. The injected pulse numbers are labeled at the low right corner and the current DW position is marked with the yellow arrows in each figure.

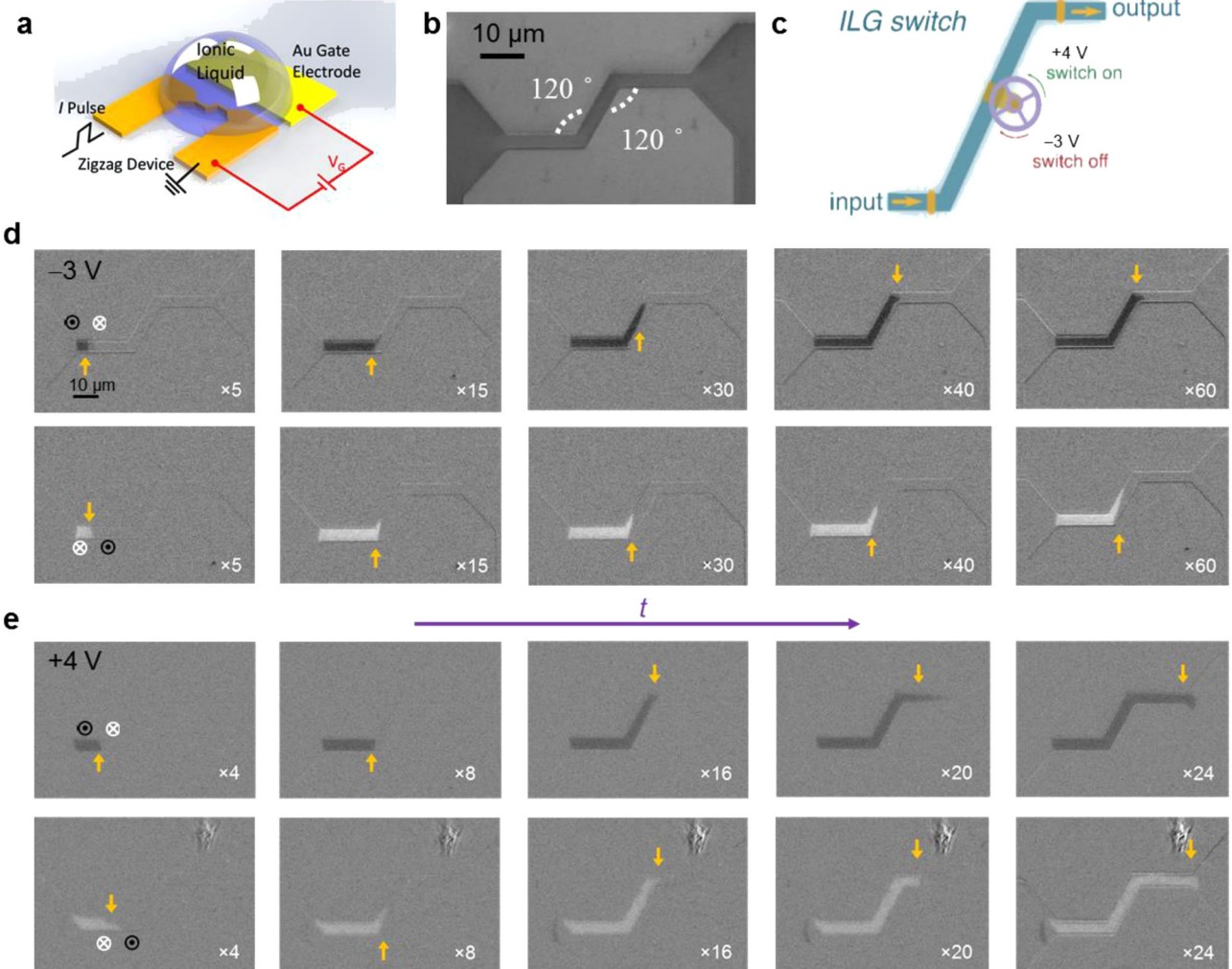

**Fig. 5 ILG-controlled logic device. a** Sketch of experimental setup for a "zigzag" device controlled by ionic liquid gating. **b** Optical micrograph of a typical zigzag device. **c** Sketch of the ILG-controlled "switch" function. Current-induced domain wall motion of a SAF zigzag device for $V_G = -3$ V (**d**) and $+4$ V (**e**). Current pulses with a magnitude and length of $1.2 \times 10^8$ A cm$^{-2}$ and 30 ns, respectively, were used. The number of injected current pulses is given in the lower right corner of each image. The state of the initial DW is schematically illustrated in the left-most image in each row of images. The DW position is marked with yellow arrows in each Kerr image.

different behavior depending on $V_G$. At $V_G = -3$ V, as shown in Fig. 4c, when a down/up DW enters the knot device and is split into two individual DWs, the DWs exhibit highly asymmetric behavior: in the lower curved lane, the DW shows a faster velocity than in the upper curved lane, and although the DW successfully passes through the lower curved lane into the straight wire region, the DW gets stuck when exiting the upper curved lane. On the other hand, at $V_G = +4$ V, the incoming down/up DW splits into two DWs, showing a very symmetric behavior as the two DWs pass through both lanes at almost the same velocity and exit both curved lanes freely (Fig. 4d). For an up/down DW, as shown in Supplementary Fig. 6, at $V_G = -3$ V, the DWs also show an asymmetric characteristic but in an inverted manner; at $V_G = +4$ V, the DW behaves exactly as the down/up case with a symmetric behavior in both lanes.

We further illustrate the potential of the large ILG effects on SAF racetracks to realize logic functions. We have designed a "zigzag" device (Fig. 5a), which consists of two 120° corners in a SAF racetrack, as shown in Fig. 5b. A "Switch" function can be realized using ionic liquid gating on this device. At $V_G = -3$ V, as shown in Fig. 5d, when the DW is injected from the left end into this device, it cannot pass through: the up/down DW gets stuck at the second corner (first row of Fig. 5d), whereas the down/up DW

gets stuck at the first corner (second row of Fig. 5d). At $V_G = 4$ V, as shown in Fig. 5e, the DW can freely pass through such a "zigzag" device no matter whether it is an up/down or down/up DW. Similar behavior is observed for injecting DW from the other end of the device as depicted in Supplementary Fig. 7. Thus, an ionic liquid gate-controlled "switch" function is realized as illustrated in Fig. 5c. After a DW is injected, when $V_G = 4$ V, such a switch is turned on, so the output of a corresponding DW is observed; when $V_G = -3$ V, the switch is turned off and no output DW is generated.

These results are consistent with the discussion above. As illustrated in the straight wire case, ILG results in the reversible transformation between high $(+4$ V$)$ and low $(-3$ V$)$ CIDWM velocities, as the nature of the DW varies between a nearly compensated chiral DW to a simple chiral DW. It has previously been shown that DWs in a perfectly compensated SAF structure can move freely regardless of the track geometry, e.g., the curvature of the wires, by current, whereas in FM structures, the DW motion is strongly affected by local geometries and the DW configuration (up/down or down/up)[26,27]. From these examples of a "knot" and a "zigzag" device (corresponding results for a deposited FM and SAF film are shown in Supplementary Figs. 5 and 8, respectively), we illustrate that by taking advantage of the large ILG-induced effects on

CIDWM in SAF structures, these effects can realize novel logic devices using suitably designed racetrack device geometries.

In summary, we find that ionic liquid gating can be used to manipulate the CIDWM in SAF structures with a significant change in DW velocity of >130 m/s. We show that the mechanism relies on changes in the exchange-coupling torque in SAF structures, which is fundamentally distinct from previous studies of FM structures[15–17]. Such changes are realized by ionic liquid gating via the manipulation of the magnetic moment of the UM sub-layer in the SAF structure, via ionic liquid gate-induced oxygen ion migration and thereby chemical oxidation and reduction of the magnetic constituents of these layers. We have also successfully demonstrated that by using such an ionic liquid gate, control of DW motion in SAF structures can allow for novel DW logic devices.

## Methods

**Sample preparation**. SAF structure of TaN(20)/Pt(15)/Co(3)/Ni(7)/Co(1.5)/Ru(5)/Co(5)/Ni(7)/Co(1.5)/TaN(15) deposited on $SiO_2$/Si substrates at room temperature by DC magnetron sputtering in a high vacuum deposition system. The deposition rates of the individual sputter sources were calibrated by an in situ quartz crystal microbalance and by ex situ X-ray reflection of individually deposited layers. For CIDWM and transport measurements, the thin films were respectively fabricated into wire structures (3 µm wide and 50 µm long) and Hall bars (effective channel: 100 µm wide and 400 µm long) with lateral Ru(50)/Au(500) gate electrodes, prepared by photolithography, Ar-ion milling, and deposition/lift-off processes. A drop of ionic liquid EMIM-TFSI [1-Ethyl-3-methylimidazolium bis(trifluoromethylsulfonyl)imide] covered the racetrack and part of the gate electrode. $V_G$ was applied between the gate and source electrodes.

**Measurement setup**. The CIDWM measurements were carried out using in situ $V_G$ application. $V_G$ was applied for 30 min before each measurement. CIDWM measurements were carried out using Kerr microscopy in a cryostat with a vacuum of ~5 × 10$^{-6}$ mbar. Magnetization and XPS measurements were carried out ex situ after removal of the ionic liquid after gating. Then, $V_G$ was set to zero and the samples were cleaned using first acetone and second ethanol before the corresponding measurement. This was possible because of the non-volatile ILG effect. Magnetization measurements were carried out in a superconducting quantum interference device-VSM. XPS measurements were carried out in a K-Alpha$^{TM}$+ system using an Al-$K\alpha$ X-ray source. The in situ transport measurements were carried out in a physical property measurement system in a few Torr He. All the measurements were carried out at room temperature.

## Data availability

The data that support the findings of this study are available from the corresponding author upon reasonable request.

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

## Acknowledgements

This project has received funding from the European Research Council (ERC) under the European Union's Horizon 2020 research and innovation program (grant agreement number 670166).

## Author contributions

S.S.P.P. conceived and directed the project. X.Z. and Y.G. prepared the samples and devices, respectively. Y.G. and F.L. carried out the transport and magnetization, and the Kerr microscopy measurements. F.L. carried out the XPS measurements. T.M. built the Kerr microscope setup for CIDWM. S.-H.Y. developed the 1D model. All authors participated in discussing the data and writing the manuscript.

## Funding

## Competing interests

The authors declare no competing interests.
