## [Peer Review File · Nature Communications]

Reviewers' Comments:

Reviewer #1:

Remarks to the Author:

Gate voltage induced domain wall velocity changes in wires is an important pathway to manipulate electrically spintronic devices at low power. Based on their previous work on synthetic antiferromagnetic (SAF) structures where DW can be moved very efficiently through Exchange Coupling Torque (ECT), the authors show clear evidence that gate voltage applied through ionic liquids can significantly affect DW velocity in SAF elements. Although this paper presents interesting experimental results toward controlling fast DW motion through reversible oxidation, I believe that it doesn't meet the requirements for publication in Nature Communication as explained below :

-Even if the effect is large, to my opinion this work is not new in the sense that both ECT induced DW motion (see Ref 10) and oxygen migration under voltage (see Ref 24,25) have been both demonstrated previously. This paper combines both effects in one single SAF device. In other words, I think this work is a very good piece of engineering research but not original enough to meet the requirements for publication in Nature Communication

-The DW velocity variation is up to 130 m/s, which is very similar to the paper cited by the authors (Ref 25) where a single ferromagnetic layer is used and oxygen migration is also induced through ionic liquids.

-The authors claim that their results originate from the full oxidation of the upper magnetic layer. However, the origin of oxidation under voltage is not discussed at all in the paper. In particular, the electric field is applied through ionic liquids in contact with an ultra-thin metallic TaN layer. The degree of oxydation of such thin TaN layer as well as the role of ambient moisture play a crucial role. Further experiments would be needed to understand the underlying effect

-The demonstration of a Knot race track device is very interesting. However, such concept could be also developed in regular ferromagnetic films where local gate can modulate DWs motion in each curved lane

Reviewer #2:

Remarks to the Author:

The paper describes experiments of current induced motion in a synthetic antiferromagnet subjected to ionic gating.

The authors show that the DW velocity is modified reversibly by the application of a gate voltage. This is consistent with the oxidation of the top FM layer, which causes the system to behave either as a simple FM layer or as a perfectly compensated SAF. The DW velocity is larger in the case of the SAF compared to the case of the simple FM.

While I find this interesting, I am not sure about the importance and novelty that it represents. It has been previously shown that the ionic gating can be used for the FM oxidation, and it has been previously shown that the SAF structure allows faster current induced motion compared to a single FM. In this work, the authors merely assemble these two previous results.

It is true that they show a large modulation of the velocity. The problem is, that in a real device, this type of oxidation would translate to a velocity drop rather than an increase compared to a well-compensated, non-oxidized SAF. The gating (+4V) improves the velocity only because in the initial state (0V) Ni and Co are a bit oxidized. It would be easier to defend their point, if the voltage gating would lead to a gain in performance, and not a loss.

In my opinion the only way that the ionic gating could be useful for applications, is to use it , as the authors suggest, as a supplementary degree of freedom for logic devices. However, they do not explain explicitly how this would function, they do not detail what level of performance it could have, nor do they build a logic gate prototype that actually works.

In my opinion, the physical phenomena presented in this paper are not sufficiently novel to interest a broad audience, but their applications could be. The paper could be significantly improved from this perspective, to the level required by the Nature journals, if the authors would

build functional logic gates based on this principle.

Concerning the technical part, I have a question and a comment:

1. Why are the velocity curves intersecting at $H_x=600$ Oe? Is this behavior reproduced by the 1D model?
2. I find that the red lines, used to indicate the position of the DWs in the "knot" device, are misleading. In the case of the SAF, where the contrast is weak, the red lines do not appear to indicate the correct DW position. Perhaps it would be better to indicate the position using arrows that do not cover the actual device, so the reader could actually see where the DWs sit.

Reviewer #3:

Remarks to the Author:

Voltage control of magnetism is of great interest due to the realization of fast, compact, and energy-efficient spintronic devices. Recently, manipulating the interlayer exchange coupling in synthetic antiferromagnetic multilayers through ionic liquid gating has become a hot topic in spintronics due to the manipulation of coupling mode non-volatile manner. Hence, the key question needs to move forward in designing energy-efficient MRAM devices through such a mechanism. This manuscript systemically studies the gating control domain wall motion in SyAF structures. As a follow-up work based on ionic liquid gating control SyAF, the significant change in DW velocity and exchange coupling torque mechanism has also been studied. The experiment characteristics are substantial, while the explanations of the results are slightly insufficient. I would suggest publishing this work in Nature communication after taking minor revisions.

(1). The physical quantities used in Fig 2(b) need to correct. "Ms" represents the saturation magnetization. We suggest the authors use M/M_s (a.u.) or Moment (emu).

(2). The authors should give more details on XPS measurement. Is the XPS analysis conducted in an in-situ or ex-situ manner? The results show apparent peaks due to the oxidization when the gating voltage changes from +4 V to -3 V. However, this critical phenomenon is observed in a broad region even though the oxidization emerges in positive voltage, a negative voltage, or the whole area.

(3) There is a visible transition of the coupling mode when applied -3 V gating voltage. The authors should strengthen the relationship between the current-induced domain wall velocity and the AFM-FM transition.

(4) The authors applied an in-plane E-field between the device and the Au electrode in this work. However, the ionic liquid will move along the E-field direction. The authors should give a detailed description on how to avoid the movement of the ionic liquid.

(5) The oxidization peaks emerge at the region from 4 V to -3 V. The electrochemical reaction mechanism may influence on the gating process. We suggest the authors add the electrochemical window test to distinguish the electrostatic doping mechanism and the electrochemical oxidation reaction mechanism.

(6) The authors conduct the ionic liquid gating control domain wall velocity based on Co/Ni perpendicular SyAF structures. Previous works have proved that liquid gating can effectively control PMA properties. The authors should give a reasonable explanation on if the changes of PMA properties would affect the DW velocity in this works.

Reviewer #1 (Remarks to the Author):

Gate voltage induced domain wall velocity changes in wires is an important pathway to manipulate electrically spintronic devices at low power. Based on their previous work on synthetic antiferromagnetic (SAF) structures where DW can be moved very efficiently through Exchange Coupling Torque (ECT), the authors show clear evidence that gate voltage applied through ionic liquids can significantly affect DW velocity in SAF elements. Although this paper presents interesting experimental results toward controlling fast DW motion through reversible oxidation, I believe that it doesn't meet the requirements for publication in Nature Communication as explained below:

-Even if the effect is large, to my opinion this work is not new in the sense that both ECT induced DW motion (see Ref 10) and oxygen migration under voltage (see Ref 24,25) have been both demonstrated previously. This paper combines both effects in one single SAF device. In other words, I think this work is a very good piece of engineering research but not original enough to meet the requirements for publication in Nature Communication

We appreciate the reviewer for his/her comment on our work as 'a very good piece of engineering research'. However, we do not agree that our work is merely the combination of known effects. The originality of our paper is as follows: We have realized the first ionic liquid gate control of *current-induced* domain wall motion while previous research focused rather on the ILG control of *field-induced* DW motion in the flow (Ref. 25) and the creep regime (Ref. 24) and, moreover only in simple, un-patterned ferromagnetic films. There exists a fundamental difference between *field-induced* and *current-induced* DW motion, as only the latter involves chiral spin orbit and giant exchange coupling torques (ECTs). Current induced DW motion is technologically by far the most important, and thus its possible modification by ILG, which had not been demonstrated before our work, is highly interesting. We have discovered that ILG modifications of these torques gives rise to very large changes in DW velocity in racetrack devices which had not been anticipated. We demonstrate that this is most important in SAF racetrack structures because of the sensitivity of the ECT to the balance in the magnetization of the upper and lower racetracks. This was not even in focus in the previous studies of the ILG of simple FM films mentioned by the reviewer. However, we do agree with the reviewer that these previous papers that showed that oxygen migration can influence the magnetic properties of ferromagnetic films which is certainly a very interesting effect. We note here that the role of oxygen (and hydrogen) in manipulating the static magnetic properties of thin films (e.g. perpendicular magnetic anisotropy) was first discussed several decades ago! Here we show how current induced magnetization dynamics can be influenced by oxygen migration controlled here via ionic liquid gating. The dramatic effects that we report are novel, unanticipated and are of fundamental interest. We believe, consequently, that this work is highly suited to Nature Communications!

Another point is that in ref. 25 the ILG changes in the field induced DW motion was irreversible whereas the large changes we report in our manuscript in the ILG current induced DW motion is non-volatile but reversible.

To emphasize how the effects that we report could be used to create novel devices we have carried out additional studies which we report in our revised manuscript that show a ILG-

controlled DW switch device (see main text (page 7, line 6 and Figure 5) and SI (Figure S7 and S8).

-The DW velocity variation is up to 130 m/s, which is very similar to the paper cited by the authors (Ref 25) where a single ferromagnetic layer is used and oxygen migration is also induced through ionic liquids.

With respect to the reviewer's comment about the change in DW velocity we would like to point out that the maximum change in the field induced DW velocity in Ref. 25 is ~50 m/s. Moreover, using higher current densities, we have observed significantly larger DW velocity variations of up to 200 m/s. This was included in Fig. 1d of our original paper. We have added a sentence in the main text to make this important point (page 3, line 18). We thank the reviewer for making this excellent point. We emphasize again that we realize the first ionic liquid gate control of *current-induced* domain wall motion while the former research focused on *field-induced* DW motion.

-The authors claim that their results originate from the full oxidation of the upper magnetic layer. However, the origin of oxidation under voltage is not discussed at all in the paper. In particular, the electric field is applied through ionic liquids in contact with an ultra-thin metallic TaN layer. The degree of oxidation of such thin TaN layer as well as the role of ambient moisture play a crucial role. Further experiments would be needed to understand the underlying effect

We thank the reviewer for this interesting question. As suggested by the reviewer, we have carried out depth profile XPS measurements of the Ta-4f spectrum in two samples with different TaN capping layer thicknesses, one 50 Å and the other 15 Å. Data is shown in Fig. R1 for the as-grown films. The Ta 4f_{7/2} and 4f_{5/2} XPS peaks shift in energy when the Ta is oxidized. And since the XPS signal originates from a finite thickness of the TaN film, for the thick TaN, one observes at the surface of the film 4 peaks, two from the TaOx and two from the TaN within the lower part of the TaN film; on the other hand, for the thin TaN, the whole layer is oxidized so the Ta XPS spectrum is dominated by the TaOx peaks. These data thus account for the origin of the oxygen which lies within the TaN layers. This oxygen migrates under ionic liquid gating into the magnetic layer from the reservoir in the TaN layer. We believe that these new XPS data, taken together with the original Co and Ni XPS data that we had included in our main paper, clearly reveal where the oxygen comes from during ILG and, indeed, further support our model that oxidation of the upper magnetic layer in the SAF structure is the main reason for the ILG effects. These new Ta XPS data have been included in both the revised main text (page 5, line 23) and SI (Figure S3).

Figure R1| Depth profile of the Ta-4f XPS spectrum for two structures with 50 Å and 15 Å thick capping layers. The stack sequence of the films used here are TaN(20)/Pt(15)/Co(3)/Ni(7)/Co(1.5)/Ru(5)/Co(5)/Ni(7)/Co(1.5)/TaN(X), all units in Å. (a) and (b) are the schematic illustrations of the oxidation for the thick (X = 50) (a) and thin (X = 15) (b) TaN capping layer cases; depth profiled Ta-4f XPS spectra are shown in (c) and (d), correspondingly. Successive spectra correspond to increasing depth into the respective films by ~ 1.2 Å.

-The demonstration of a Knot race track device is very interesting. However, such concept could be also developed in regular ferromagnetic films where local gate can modulate DWs motion in each curved lane

We appreciate the reviewer for his/her interest in our knot device. The function that we show in our paper is only possible when the SAF layer is gated (by oxidation) to a FM layer or vice versa. We switch between a knot device in which both lanes behave the same (SAF) to a device in which the two lanes are distinct (FM). The performance of this knot device depends on the fact that the DW is chiral. We have further utilized this effect to build functional logic device. These new results, based on the ILG gate induced transition between a SAF and a simple FM layer, are illustrated in our revised main text and in our updated SI Fig. S7 and S8.

Reviewer #2 (Remarks to the Author):

The paper describes experiments of current induced motion in a synthetic antiferromagnet subjected to ionic gating.

The authors show that the DW velocity is modified reversibly by the application of a gate voltage. This is consistent with the oxidation of the top FM layer, which causes the system to behave either as a simple FM layer or as a perfectly compensated SAF. The DW velocity is larger in the case of the SAF compared to the case of the simple FM.

While I find this interesting, I am not sure about the importance and novelty that it represents.

It has been previously shown that the ionic gating can be used for the FM oxidation, and it has been previously shown that the SAF structure allows faster current induced motion compared to a single FM. In this work, the authors merely assemble these two previous results.

It is true that they show a large modulation of the velocity. The problem is, that in a real device, this type of oxidation would translate to a velocity drop rather than an increase compared to a well-compensated, non-oxidized SAF. The gating (+4V) improves the velocity only because in the initial state (0V) Ni and Co are a bit oxidized. It would be easier to defend their point, if the voltage gating would lead to a gain in performance, and not a loss.

In my opinion the only way that the ionic gating could be useful for applications, is to use it, as the authors suggest, as a supplementary degree of freedom for logic devices. However, they do not explain explicitly how this would function, they do not detail what level of performance it could have, nor do they build a logic gate prototype that actually works.

In my opinion, the physical phenomena presented in this paper are not sufficiently novel to interest a broad audience, but their applications could be. The paper could be significantly improved from this perspective, to the level required by the Nature journals, if the authors would build functional logic gates based on this principle.

We thank the reviewer for his/her comments. Based on our former results, we further implement the ionic liquid gating together with a designed 'zigzag' device to realize an ILG-controlled 'switch' function. These new results are included in the both our revised Main text (page 7, line 6 and Figure 5) as well as our updated SI (Figure S7 and S8). By demonstrating such an example of an ionitronic-based spintronic switch as a component of a potential logic technology, we believe our work creates a new platform for memory-in-logic applications.

Concerning the technical part, I have a question and a comment:

1. Why are the velocity curves intersecting at $H_x=600$ Oe? Is this behavior reproduced by the 1D model?

We appreciate the reviewer for his/her carefully reading our manuscript and this interesting question. There is no particular reason why the curves intersect at $H_x=600$ Oe.

2. I find that the red lines, used to indicate the position of the DWs in the "knot" device, are misleading. In the case of the SAF, where the contrast is weak, the red lines do not appear to indicate the correct DW position. Perhaps it would be better to indicate the position using arrows that do not cover the actual device, so the reader could actually see where the DWs sit.

We thank the reviewer for his/her advice. We have changed the red lines into yellow arrows that do not cover the actual device as the reviewer suggests in both Fig. 4 in main text and in Figs. S5 and S6 in SI.

Reviewer #3 (Remarks to the Author):

Voltage control of magnetism is of great interest due to the realization of fast, compact, and energy-efficient spintronic devices. Recently, manipulating the interlayer exchange coupling in synthetic antiferromagnetic multilayers through ionic liquid gating has become a hot topic in spintronics due to the manipulation of coupling mode non-volatile manner. Hence, the key question needs to move forward in designing energy-efficient MRAM devices through such a mechanism. This manuscript systemically studies the gating control domain wall motion in SyAF structures. As a follow-up work based on ionic liquid gating control SyAF, the significant change in DW velocity and exchange coupling torque mechanism has also been studied. The experiment characteristics are substantial, while the explanations of the results are slightly insufficient. I would suggest publishing this work in Nature communication after taking minor revisions.

(1). The physical quantities used in Fig 2(b) need to correct. "Ms" represents the saturation magnetization. We suggest the authors use M/Ms (a.u.) or Moment (emu).

We appreciate the reviewer's kind reminder. In our updated Figure 2b, the quantity 'M/Ms (a.u.)' is used.

(2). The authors should give more details on XPS measurement. Is the XPS analysis conducted in an in-situ or ex-situ manner? The results show apparent peaks due to the oxidization when the gating voltage changes from +4 V to -3 V. However, this critical phenomenon is observed in a broad region even though the oxidization emerges in positive voltage, a negative voltage, or the whole area.

We thank the reviewer very much for the comment. The XPS analysis is conducted *ex-situ*: we have added this point in the revised main text (page 5 line 16) and the Methods section (page 9, line 16). With respect to the second point, we have addressed this point in our response to Reviewer 1 above.

(3). There is a visible transition of the coupling mode when applied -3 V gating voltage. The authors should strengthen the relationship between the current-induced domain wall velocity and the AFM-FM transition.

We appreciate the reviewer for his/her comment. We have made some changes in the manuscript to make the point clearer that ILG causes a change from a SAF to an FM structure.

(4). The authors applied an in-plane E-field between the device and the Au electrode in this work. However, the ionic liquid will move along the E-field direction. The authors should give a detailed description on how to avoid the movement of the ionic liquid.

We appreciate the reviewer's comment. However, in our experiment, we did not observe any movement of the ionic liquid during the gating process. One possible explanation is that in our experiments, the region of interest (100 μm \times 100 μm) is in the middle of the liquid and is

much smaller than the area covered by the liquid (2 mm × 2 mm) so any movement of the liquid would not be expected to influence the measurements.

(5). The oxidization peaks emerge at the region from 4 V to -3 V. The electrochemical reaction mechanism may influence on the gating process. We suggest the authors add the electrochemical window test to distinguish the electrostatic doping mechanism and the electrochemical oxidation reaction mechanism.

We thank the reviewer for his/her comment. While this is an interesting point, we find no evidence for pure electrostatic effects but rather only for electrochemical effects in our work. Moreover, these effects are very sensitive to the rate at which the voltage is swept, as is common for electrochemical effects. Our manuscript concerns these electrochemical effects which lead to the non-volatile changes that we report.

(6) The authors conduct the ionic liquid gating control domain wall velocity based on Co/Ni perpendicular SyAF structures. Previous works have proved that liquid gating can effectively control PMA properties. The authors should give a reasonable explanation on if the changes of PMA properties would affect the DW velocity in this works.

We thank the reviewer for his/her comment. As suggested by the reviewer, the uniaxial magnetic anisotropy energy could also be affected by ILG. In this work, however, the lower magnetic layer in the SAF structure is the dominant source of the PMA: it arises from the interface between this layer and the Pt underlayer. The ILG process, that we have demonstrated, is largely a consequence of migration of oxygen from the TaN capping layer, will predominantly affect the upper magnetic layer. Thus, we don't expect significant changes in the uniaxial magnetic anisotropy energy.

Reviewers' Comments:

Reviewer #1:

Remarks to the Author:

My feeling is that the authors have answered all the questions and the manuscript meets now the requirements for publication in Nature communication

Reviewer #2:

None

Reviewer #3:

Remarks to the Author:

Voltage control of magnetism is of great interest due to the realization of fast, compact, and energy-efficient spintronic devices. Recently, manipulating the interlayer exchange coupling in synthetic antiferromagnetic multilayers through ionic liquid gating has become a hot topic in spintronics due to the manipulation of coupling mode non-volatile manner. Hence, the key question needs to move forward in designing energy-efficient MRAM devices through such a mechanism. This manuscript systemically studies the gating control domain wall motion in SyAF structures. As a follow-up work based on ionic liquid gating control SyAF, the significant change in DW velocity and exchange coupling torque mechanism has also been studied. The experiment characteristics are substantial, while the explanations of the results are slightly insufficient. I would suggest publishing this work in Nature communication after taking minor revisions.

(1). The physical quantities used in Fig 2(b) need to correct. "Ms" represents the saturation magnetization. We suggest the authors use M/M_s (a.u.) or Moment (emu).

(2). The authors should give more details on XPS measurement. Is the XPS analysis conducted in an in-situ or ex-situ manner? The results show apparent peaks due to the oxidization when the gating voltage changes from +4 V to -3 V. However, this critical phenomenon is observed in a broad region even though the oxidization emerges in positive voltage, a negative voltage, or the whole area.

(3) There is a visible transition of the coupling mode when applied -3 V gating voltage. The authors should strengthen the relationship between the current-induced domain wall velocity and the AFM-FM transition.

(4) The authors applied an in-plane E-field between the device and the Au electrode in this work. However, the ionic liquid will move along the E-field direction. The authors should give a detailed description on how to avoid the movement of the ionic liquid.

(5) The oxidization peaks emerge at the region from 4 V to -3 V. The electrochemical reaction mechanism may influence on the gating process. We suggest the authors add the electrochemical window test to distinguish the electrostatic doping mechanism and the electrochemical oxidation reaction mechanism.

(6) The authors conduct the ionic liquid gating control domain wall velocity based on Co/Ni perpendicular SyAF structures. Previous works have proved that liquid gating can effectively control PMA properties. The authors should give a reasonable explanation on if the changes of PMA properties would affect the DW velocity in this works.

Reviewer #1 (Remarks to the Author):

My feeling is that the authors have answered all the questions and the manuscript meets now the requirements for publication in Nature communication

We thank the reviewer for his/her recommendation to publish our paper in Nature Communications.

Reviewer #3 (Remarks to the Author):

The reviewer recommends publication of our paper in Nature Communications. We do note that these comments are identical to those that we previously responded so I include our previous responses here (I am highlighting these in yellow since they are the same as in our previous response!).

Voltage control of magnetism is of great interest due to the realization of fast, compact, and energy-efficient spintronic devices. Recently, manipulating the interlayer exchange coupling in synthetic antiferromagnetic multilayers through ionic liquid gating has become a hot topic in spintronics due to the manipulation of coupling mode non-volatile manner. Hence, the key question needs to move forward in designing energy-efficient MRAM devices through such a mechanism. This manuscript systemically studies the gating control domain wall motion in SyAF structures. As a follow-up work based on ionic liquid gating control SyAF, the significant change in DW velocity and exchange coupling torque mechanism has also been studied. The experiment characteristics are substantial, while the explanations of the results are slightly insufficient. I would suggest publishing this work in Nature communication after taking minor revisions.

We thank the reviewer for his/her recommendation to publish our paper in Nature Communications after some minor revisions.

(1). The physical quantities used in Fig 2(b) need to correct. "Ms" represents the saturation magnetization. We suggest the authors use M/Ms (a.u.) or Moment (emu).

We appreciate the reviewer's comment. In our updated Figure 2b, the quantity 'M/Ms (a.u.)' is used.

(2). The authors should give more details on XPS measurement. Is the XPS analysis conducted in an in-situ or ex-situ manner? The results show apparent peaks due to the oxidization when the gating voltage changes from +4 V to -3 V. However, this critical phenomenon is observed in a broad region even though the oxidization emerges in positive voltage, a negative voltage, or the whole area.

We thank the reviewer for this comment: we are happy to add more details about our XPS measurements. The XPS studies are conducted *ex-situ*: we have added this point in the revised main text (page 5 line 16) and in the Methods section (page 9, line 16). With respect to the second point, we have addressed this point in our revised manuscript in both main text (page 5 line 22) and the SI (Fig. S3).

(3). There is a visible transition of the coupling mode when applied -3 V gating voltage. The authors should strengthen the relationship between the current-induced domain wall velocity and the AFM-FM transition.

We appreciate the reviewer for his/her comment. We have made some changes in the manuscript to make the point clearer that ILG causes a change from a SAF to an FM structure and that we have further utilized this phenomenon to realize a 'switch' function (page 7, line 5).

(4). The authors applied an in-plane E-field between the device and the Au electrode in this work. However, the ionic liquid will move along the E-field direction. The authors should give a detailed description on how to avoid the movement of the ionic liquid.

We appreciate the reviewer's comment. However, in our experiment, we did not observe any movement of the ionic liquid during the gating process. One possible explanation is that in our experiments, the region of interest ($100\ \mu\text{m} \times 100\ \mu\text{m}$) is in the middle of the liquid and is much smaller than the area covered by the liquid ($2\ \text{mm} \times 2\ \text{mm}$) so any movement of the liquid would not be expected to influence the measurements.

(5). The oxidization peaks emerge at the region from 4 V to -3 V. The electrochemical reaction mechanism may influence on the gating process. We suggest the authors add the electrochemical window test to distinguish the electrostatic doping mechanism and the electrochemical oxidation reaction mechanism.

We thank the reviewer for his/her comment. While this is an interesting point, we find no evidence for pure electrostatic effects but rather only for electrochemical effects in our work. Moreover, these effects are very sensitive to the rate at which the voltage is swept, as is common for electrochemical effects. Our manuscript concerns these electrochemical effects which lead to the non-volatile changes that we report.

(6). The authors conduct the ionic liquid gating control domain wall velocity based on Co/Ni perpendicular SyAF structures. Previous works have proved that liquid gating can effectively control PMA properties. The authors should give a reasonable explanation on if the changes of PMA properties would affect the DW velocity in this works.

We thank the reviewer for his/her comment. As suggested by the reviewer, the uniaxial magnetic anisotropy energy could also be affected by ILG. In this work, however, the lower magnetic layer in the SAF structure is the dominant source of the PMA: it arises from the interface between this layer and the Pt underlayer. The ILG process, that we have demonstrated, is largely a consequence of migration of oxygen from the TaN capping layer,

that will predominantly affect the upper magnetic layer. Thus, we don't expect significant changes in the uniaxial magnetic anisotropy energy.